# Biomass-Derived N-Doped Activated Carbon from Eucalyptus Leaves as an Efficient Supercapacitor Electrode Material

Dinesh Bejjanki [1], Praveen Banothu [1], Vijay Bhooshan Kumar [2,*] and Puttapati Sampath Kumar [1,*]

[1]  Department of Chemical Engineering, National Institute of Technology Warangal, Hanamkonda 506004, Telangana, India

[2]  The Shmunis School of Biomedicine and Cancer Research, George S. Wise Faculty of Life Sciences, Tel Aviv University, Tel Aviv 6997801, Israel

*  Correspondence: vijaybooshan@mail.tau.ac.il or vijaybhushan86@gmail.com (V.B.K.); pskr@nitw.ac.in (P.S.K.)

**Abstract:** Biomass-derived activated carbon is one of the promising electrode materials in supercapacitor applications. In this work bio-waste (oil extracted from eucalyptus leaves) was used as a carbon precursor to synthesize carbon material with $ZnCl_2$ as a chemical activating agent and activated carbon was synthesized at various temperatures ranging from 400 to 800 °C. The activated carbon at 700 °C showed a surface area of 1027 $m^2 g^{-1}$ and a specific capacitance of 196 $F g^{-1}$. In order to enhance the performance, activated carbon was doped with nitrogen-rich urea at a temperature of 700 °C. The obtained activated carbon and N-doped activated carbon was characterized by phase and crystal structural using (XRD and Raman), morphological using (SEM), and compositional analysis using (FTIR). The electrochemical measurements of carbon samples were evaluated using an electrochemical instrument and NAC-700 °C exhibited a specific capacitance of 258 $F g^{-1}$ at a scan rate of 5 $mV s^{-1}$ with a surface area of 1042 $m^2 g^{-1}$. Thus, surface area and functionalizing the groups with nitrogen showed better performance and it can be used as an electrode material for supercapacitor cell applications.

**Keywords:** eucalyptus leaves; activated carbon; nitrogen doping; chemical activation; supercapacitor

## 1. Introduction

Energy is necessary in day today life. In future, energy generation with sufficient and sustainable methods will be a major concern. Currently, over 75% of the energy used by society is produced by fossil fuels, including coal, natural gas, and oil, neither of which are renewable. As a result, energy must always be replaced by renewable energy before fossil fuels run out. Excessive use of fossil fuels leads to global warming. It releases harmful gases directly into the environment [1,2]. To address the challenges related to environmental pollution and energy production, fuels with petroleum must be transformed to renewable and sustainable energy sources. Unutilized waste materials and biomass can be used effectively in synthesizing of carbon materials for energy storage and conversion devices. The carbonization process carried out by heating biomasses under inert gas condition and high temperature. In contrast, the heteroatoms (oxygen, sluphur and nitrogen etc.,) in the networks of the biological macromolecules escapes there by leaving the carbon skeletons with the porous shape. So when residual carbon skeletons are activated, they may form linked 3D structures with considerably high conductivity, porosity and surface area, making them attractive candidates in energy storage applications. There are various biomass-derived carbon materials using highly effective green energy storage devices. Finding alternative energy resources that use green energy (biomass) and clean energy is essential [3,4]. So far, several storage technologies have been developed such as capacitors, batteries, fuel cells, and supercapacitors [1,2,5]. Supercapacitors are in the field of electrochemical devices with their remarkable fast charging speed, high power density, light weight, safe operation, and long life cycle [3,4,6]. Supercapacitors are divided into symmetric and asymmetric supercapacitors [5,7]. Electrochemical double-layer

capacitors (EDLC) and pseudo capacitors are classified as symmetric supercapacitors, while the hybrid capacitor is an asymmetric supercapacitor. The charge storage mechanism classifies the behavior of supercapacitor. The EDLC stores energy electrostatically while pseudo capacitors store energy via electrochemical redox or faradaic reactions. The Supercapacitor consists of 4 major components they are electrolyte, electrode, current collector, and separator. Among all the components, the performance of supercapacitor is mainly based on the electrode material [8]. Activated carbon, carbon nanotubes, and carbon nanofibers are utilized as electrode materials in EDLC, whereas metal-based oxides ($RuO_2$, $Co_3O_4$, $MnO_2$, NiO) and conducting polymers (polyaniline, poly-(3,4-ethylenedioxythiophene), and polypyrrole) are employed as pseudo capacitor materials, whereas the pseudocapacitors are chemical stability, high conductivity, corrosion resistance, controlled pore structure, temperature stability, environmental friendliness, and low cost [9].

In recent years, biomass-derived carbon material as electrode material gained a wide variety of attention, because of its unique properties such as high specific surface area (SSA), excellent electrical conductivity, and low production cost. Carbon materials exhibited in different morphologies, such as graphene, carbon nanotubes (CNT), carbon sphere, and carbon nanoparticle, thus have been looked into supercapacitor applications. Although, these electrode materials show short-time durability, low energy density, and high power density with less porosity [4,10,11]. The biomass-derived activated carbon has been widely used as electrode material for supercapacitor application. Considering the importance of activated carbon from biomass in an energy storage device, the literature has been carried out to know the chemical composition of various biomass such as Noeli et al. have studied the physical characteristics of dried bananas leaving and it has a carbon content of 43.5 wt.%. [12], D. Pujol et al. reported the elemental analysis of coffee waste extracted from soluble coffee industry with a carbon wt.% has 57 [13], Salwan et al. studied the chemical composition of black tea waste and algae both biomass waste has a carbon wt. percent of 30 & 28 respectively [14], and Yang Liu et al. studied biomass waste of willow leave, and which has 45 wt.% of carbon [15], Nannan et al. and Saad A et al. studied waste biomass, i.e., tremella also known as white fungus, and Egyptin mango leaves both biomass have the carbon as wt.% 40.25 and 40.7, respectively [16,17], and Grima et al., studied the carbonaceous residue of eucalyptus leaves biomass, and it has 74.5 wt.% carbon [18]. Compared to all other biomasses, the eucalyptus has the highest carbon content (approximately 75%) [18]. The chemical activation technique used to synthesize activated carbon will help to lower its issues and can increase its active sites. The chemical activating agent are $ZnCl_2$, KOH, $H_3PO_4$, NaOH, $K_2CO_3$, $H_2SO_4$, NaCl, and $CaCl_2$ [19–21]. The $ZnCl_2$ has been used as activating agent to prepare activated carbon mostly for lignocellulosic biomass due which acts as dehydrating and dampening agent during the chemical activation. The $ZnCl_2$ activation causes swelling in the cellulose structure due to electrolytic action which also leads to increasing the surface area of activated carbon. The $ZnCl_2$ activated carbon (AC) has been widely used as electrode material in energy storage application. Furthermore, AC having low graphitization degrees usually have poor electric conductivity, which significantly limits the quick charge and discharge, especially at higher current densities. It's been proven that modifying AC with heteroatom species such as (oxygen, sulphur, and nitrogen functional groups), not only alter the conductivity of carbon network, but also helps in the permeability ions into electrode and electrolytes.

In this study, the chemical activation technique has been employed to synthesize activated carbon, using the carbonation method at various temperatures from 400 °C to 800 °C using $ZnCl_2$ as an activating agent. As prepared activated carbon was studied to determine the effect of temperature on its morphology and specific capacitance. Among all the AC, AC-700 °C has shown the most promising results in terms of phase, morphology, structure, composition, and electrochemical measurements. In order to synthesize N-doped activated carbon, urea was used as a nitrogen precursor and a chemical activation process was applied.

## 2. Materials and Methods

### 2.1. Materials

Raw materials required to synthesis activated carbon are eucalyptus leaves, zinc chloride ($ZnCl_2$), hydrochloric acid 35% (HCl), ethanol (>99% purity), distilled water (DI), and urea ($CH_4N_2O$) reagents were purchased from Merck India Pvt Ltd. (Singapore). Nickel foam has current collector with (0.5mm $\times$ 20cm $\times$ 30 cm) dimensions purchased from (Global nanotech Pvt. Ltd., Gujarat, India).

### 2.2. Synthesis of Activated Carbon

Eucalyptus leaves were collected from the surrounding of NIT Warangal, India. Eucalyptus leaves were pre-treated with DI water and dried, oil was extracted from the eucalyptus leaves using steam distillation. After extracting oil, the left over bio-waste (eucalyptus residue) was dried at 80 °C for 8 h. The dried biomass was crushed in fine powder by using a mortar and pestle and sieved with a mesh size of 60 to 80. The resultant eucalyptus leaves powder (ELP) was carbonized in tubular furnace under an inert atmosphere (nitrogen) with an inlet flow rate of 75 mL $min^{-1}$. The furnace consist of PID temperature controller with long tube of length 160 cm and the diameter of 40 mm. Inside the tubular furnace a quartz boat is used to hold the sample. The ELP was carbonized at 300 °C for 1 h. @ 5 °C $min^{-1}$. Next, the samples were mixed with $ZnCl_2$ in the proportion of 1:4 ($ZnCl_2$: sample). The obtained mixture was agitated to create a homogenous slurry and dried for 12 h and at 100 °C. The resultant products were then carbonized to 400–800 °C for 1 h. @ 5 °C $min^{-1}$ under an inert atmosphere. The product was dried at 80 °C for 10 h. After the process was completed, the material was cooled and taken from the tubular furnace. The carbonized material was then rinsed with 1 M HCl till the pH reached 7 and then filtered and dried for 12 h at 100 °C. The resultant product of this post-treatment was activated carbon, abbreviated as AC. The AC was labelled as AC-x, where x (x = 400 to 800 °C) represents the carbonization temperature and stored AC-x materials in sample bottles.

### 2.3. Synthesis of N-Doped Activated Carbon

As obtained eucalyptus leaves powder (ELP), 10 g was weighed and carbonized at 300 °C @ 5 °C $min^{-1}$ under inert atmosphere. As obtained sample from the previous step was mixed with $ZnCl_2$ and urea in the proportion of 1:4:2 ($ZnCl_2$: sample: urea). Next, the product was then carbonized to 700 °C @ 5 °C $min^{-1}$ for 1 h. The resultant product was washed with 1 M HCl followed by DI water until the pH attained to 7 and finally the product was dried and collected. The Eucalyptus leaves N-doped carbon materials (ELNAC) so the resultant was labelled as NAC-x, where x (x = 700 °C) represents the carbonization temperature.

### 2.4. Charaterization and Electrochemical Performance Evaluations

The crystal structure of AC and NAC composition was determined using a X-ray diffraction pattern (XRD PANalytical, X'Pert-PRO MPD using Cu K$\alpha$ radiation) at a wavelength of 0.15406 nm with an angle $\theta$ ranging from 10° to 70°. The scanning step size was 0.02°, and also the rate of scanning was 0.1 s per step. Raman spectroscopy (Laser Raman Spectrometer, China). Raman spectra were recorded using 523 nm laser source, at wavelength ranging from 500 to 2700 $cm^{-1}$. The different functional groups and vibrational modes associated with the (AC and NAC) material were studied using FTIR analysis in the range of 400 to 4000 $cm^{-1}$. The SEM and Raman spectroscopy analysis were carried out to understand the morphology of the material and to know to the presents of defects in the (AC and NAC).

The electrochemical performance was evaluate using three-electrode system CH instrument (model-CHI660D) in aqueous 2M KOH electrolyte. In this configuration, carbon electrodes, platinum (Pt), and Ag/AgCl, were employed as the working, counter, and reference electrodes, respectively. The working electrode concocted by means of active material (AC's and NAC), conducting graphite, and polyvinylidene fluoride (PVDF) were

mixed at a weight percentage of 80:10:10, and N-Methyl-2-pyrrolidone used as a solvent to make a homogeneous slurry. After that, the obtained slurry was coated on a 1*2 cm$^2$ nickel foam and then dried at 85 °C for 4 h. As prepared working electrode used to evaluate the performance curve of cyclic voltammetry (CV), and galvanostatic charge-discharge (GCD) at different scan rates and current densities in the potential range of −1.0 to 0 V. The properties of the electrode such as specific capacitance, energy density, and power density can be calculated from the GCD curves using the below Equations (1)–(3), respectively.

$$C = \frac{I \Delta t}{m \Delta V} \tag{1}$$

$$E = \frac{C \, \Delta V2}{2} \tag{2}$$

$$P = \frac{E}{\Delta t} \tag{3}$$

where, $C$ = specific capacitance (F g$^{-1}$), $\Delta t$ = discharge time difference (s), $I$ = current (mA), $m$ = mass of active material in working electrode (mg), $\Delta V$ = the discharge voltage window (V), $E$ = energy density (Wh kg$^{-1}$), P = power density (W kg$^{-1}$).

### 3. Results

*3.1. X-ray Diffraction Spectroscopy*

The XRD pattern of AC at various temperatures (400 °C to 800 °C) and NAC at 700 °C was characterized as seen in Figure 1a,b. The XRD pattern of AC and NAC exhibits two diffraction peaks at 2θ values of 24° and 44° which correlate to the miller indices of (002) and (100), respectively. The obtained peak was broad shows that an amorphous carbon structure with a low degree of graphitization [22,23]. This result shows that as we increase the carbonization temperature leads to the carbon material becomes more graphitic and increases the conductivity of a sample [24].

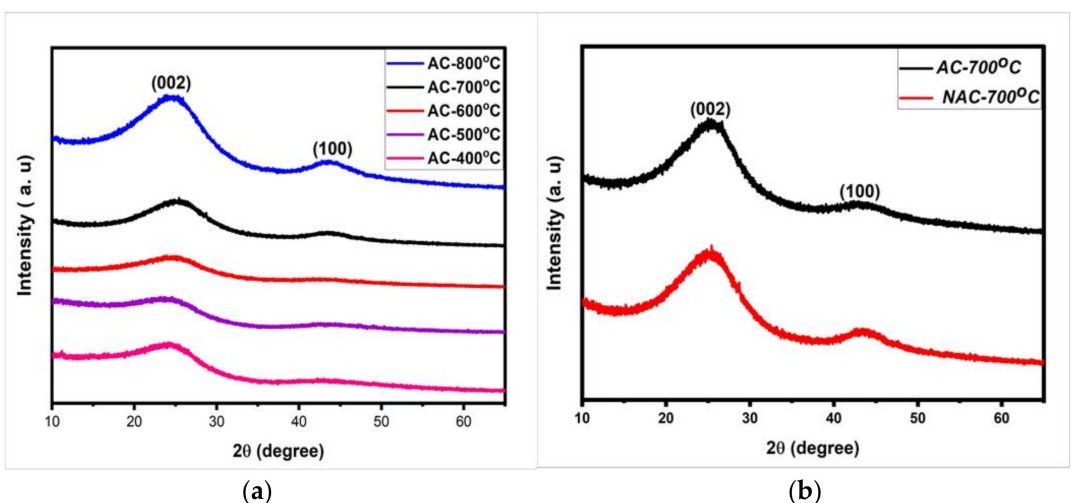

**Figure 1.** XRD pattern of (**a**) various temperature AC (**b**) AC-700 °C and NAC-700 °C.

Table 1 Represents the interlayer d-spacing and dimensions of microcrystalline of various temperature carbon materials. Corresponding to Table 1, the increase in activation temperature guides to a change in the stack height (L$_c$) and microcrystalline structure. The stack height (L$_c$) is related to the specific surface area provided by empirical formula (SSA$_{xrd}$ = 2/ρ$_{xrd}$ L$_c$), whereas ρ$_{xrd}$ is calculated from ρ$_{xrd}$= (d$_{002(graphite)}$/d$_{002}$) × ρ$_{graphite}$ and (L$_a$) is stack width. Based on the SSA$_{xrd}$ formula, the stack height (L$_c$) and surface area are inversely proportional. If the L$_c$ value is low, it means that the surface area from XRD is high and vice versa. From Table 1, AC-700 has the least stack height (L$_c$) of 9.39 °A, which

shows that AC-700 has the highest specific surface area compared to various temperature electrodes [25,26].

**Table 1.** Interlayer d-spacing and microcrystalline dimensions of carbon material at various temperatures. (* is corresponding to the XRD major peak).

| Sample Code | 2θ | | Inter Layer (nm) | | Micro Crystalline Dimension | | $L_c/L_a$ | Np | $SSA_{xrd}$ |
|---|---|---|---|---|---|---|---|---|---|
| | * C002 | * C100 | d002 | d100 | $L_c$ | $L_a$ | | | |
| AC-800 | 24.35 | 43.72 | 0.365 | 0.2067 | 0.957 | 3.506 | 0.273 | 2.621 | 1008.34 |
| AC-700 | 24.85 | 43.87 | 0.357 | 0.2061 | 0.939 | 3.006 | 0.312 | 2.625 | 1027.74 |
| AC-600 | 24.26 | 43.71 | 0.366 | 0.2068 | 0.989 | 2.626 | 0.376 | 2.699 | 975.73 |
| AC-500 | 23.92 | 43.96 | 0.371 | 0.2072 | 1.000 | 2.156 | 0.463 | 2.692 | 965.10 |
| AC-400 | 23.85 | 43.01 | 0.372 | 0.2100 | 1.024 | 2.854 | 0.358 | 2.748 | 942.54 |
| NAC-700 | 25.05 | 43.49 | 0.354 | 0.2078 | 0.925 | 2.917 | 0.317 | 2.607 | 1042.92 |

### 3.2. Raman Spectroscopy

The Raman spectroscopy technique is commonly used to characterize carbon materials due to its ability to reveal the disordered structure of carbon molecules. The Raman spectrum of AC-700 °C and NAC-700 °C exhibits two peaks that corresponds to the D-band at (1360 cm$^{-1}$) and G-band at (1585 cm$^{-1}$) while the D-band represent the disorder in structure, and G-band represents vibration of carbon atoms, correspondingly as seen in (Figure 2). The intensity ratio of the D and G band is utilized to assess the level of graphitization in AC and NAC [23]. The intensity ratio ($I_D/I_G$) of AC and NAC are 0.9995 and 1.05, respectively. It indicates that the effect of N-doping guides the degree of the disorder and incorporation of heterogeneous atom (nitrogen) into the graphite layer therefore it can relate to, higher the intensity ratio the higher is the degree of disorder [27].

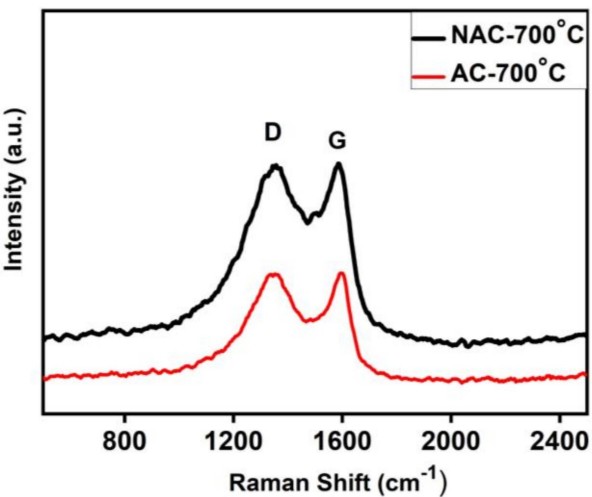

**Figure 2.** Raman spectroscopy of AC-700 °C and NAC-700 °C.

### 3.3. Scanning Electronic Microscopy (SEM)

The morphological structures and characteristics of untreated, AC (at different carbonization temperatures), and NAC were analyzed by using the SEM technique, and Figure 3b–f shows the microstructure of AC samples of temperature from 400 °C to 800 °C at different magnification. The observation of micrographs (SEM Image) having good porosity with macro pores in nature has resulted from the synthesized AC. As can be seen from Figure 3a, there are some vacant sites after pre-carbonization (untreated). Once it is

activated with chemical activating agent guides to activate the inactive sites and as increasing the carbonization temperature leads to more active site formation so that the porosity of AC has been increased which corresponds to increasing the specific surface area of AC [26]. As observed from Figure 3e,g, most of the inactive site are active as carbonization temperature reaches to 700 °C. From Figure 3f, as the temperature increase further lead to damaging pores as well as non-uniform porous material. Further, the chemical composition of AC at different carbonization temperatures and NAC was determined using energy dispersive spectroscopy (EDS). A summary of the chemical composition of the different samples is presented in Figure S1 and Table S1. According to Table S1, the weight and atomic percentage of carbon increase as the carbonization temperature increases. Low to high heating rates increased the carbon content of the AC from 80.2 to 97.5 wt.%, whereas low to high heating rates decreased the oxygen content from 15.0 to 0%. Therefore, the formation of porosities in AC facilitates the mobility of ions between both the electrode and electrolyte during the charge and discharge cycle [28].

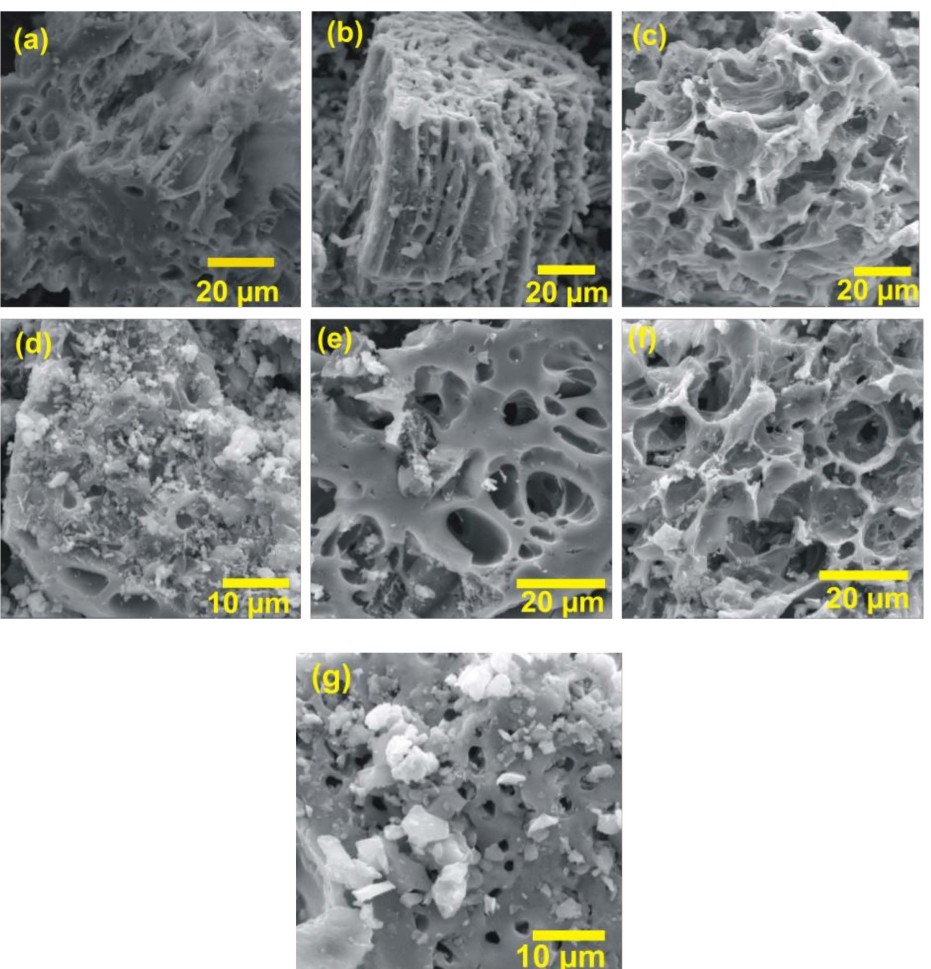

**Figure 3.** SEM images of (**a**) untreated, (**b**) AC-400 °C, (**c**) AC-500 °C, (**d**) AC-600 °C, (**e**) AC-700 °C, (**f**) AC-800 °C, and (**g**) NAC-700 °C.

### 3.4. Fourier-Transform Infrared Spectroscopy (FTIR)

The FTIR analysis of AC at a carbonization temperature of 700 °C was analyzed using the FTIR spectrometer within the scale of 400–4000 cm$^{-1}$. Figure 4a shows The sharp band around 3650 cm$^{-1}$ was observed and represents the alcohol groups of O-H stretching vibration [29]. The small band were appearing at 3135 cm$^{-1}$ and 2895 cm$^{-1}$ correspond to the alcohol groups of O-H stretching of weak and intramolecular bonded. The strong band was appearing at 1725 cm$^{-1}$ and was generally ascribed to aldehyde groups of C=O

stretching vibration [30]. The strong peak at 1530 cm$^{-1}$ was appearing and represent the nitro compound of N-O stretching vibration. The medium band at 1325 cm$^{-1}$ was observed and related to alcohol groups of O-H bending alcohol. The strong peak at 1110 cm$^{-1}$ was ascribed to the aliphatic ether of C-O stretching vibration. The two strong bands at 885 and 695 cm$^{-1}$ appearing may be related to alkene groups of C=C bending vibration [29].

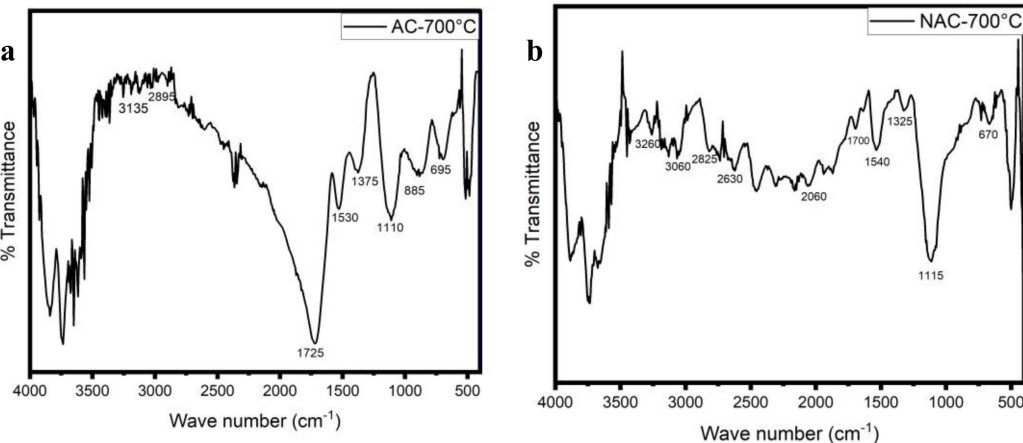

**Figure 4.** FTIR spectra of (**a**) AC-700°C and (**b**) NAC-700°C.

Figure 4b show the FTIR spectrum NAC-700 °C of the peak at 3260 cm$^{-1}$ was observed as a weak band and represents an alcohol group of O-H stretching vibration. The two small peaks around 3060 and 2825 cm$^{-1}$ were assigned to the aliphatic group CH, -CH$_2$, and –CH$_3$ [29,31]. The band at 2630 cm-1 was observed and assigned to aldehyde groups of C-H stretching. The weak band appearing at 1700 cm$^{-1}$ was allotted to carbonyl groups of C=O [29,32]. The strong peaks at 1540 cm$^{-1}$ and attributed to the nitro compound group of N-O stretching vibrations. The weak band at 1325 cm$^{-1}$ was generally credited to the alcohol group of O-H bending vibration. The strong peak at 1110 cm$^{-1}$ was ascribed to the aliphatic ether of C-O stretching vibration. The band at 670 cm$^{-1}$ may be ascribed to alkene groups of C=C bending vibration [29].

*3.5. Cyclic Voltammetry (CV)*

Evaluation of the capacitive performance of activated carbon prepared at various carbonization temperatures as electrode material for supercapacitor device was performed with cyclic voltammetry (CV) to study the result of scan rate on specific capacitance, using different scan rates (5–100 mV s$^{-1}$), and voltage windows of −1 to 0 V. As observed from Figure 5a shows EDLC behavior of AC and NAC as similar as ref [33], it is clearly seen the AC and NAC at 700 °C with 5 mV s$^{-1}$ shows a high enclosed area. In Figure 5b,c, The AC and NAC with the slight disorder and increasing the scan rate can change the area enclosed within the curve but decrease the specific capacitance value [23,25]. The best specific capacitance (196 F g$^{-1}$) resulted for AC-700 °C at the scan rate of 5 mV s$^{-1}$ in comparison to all other carbonization temperatures and NAC-700 °C has shown better specific capacitance (252 F g$^{-1}$) comparing to AC-700 °C from CV curves. Figure 5d shows that increasing the carbonization temperature guide to increasing the specific capacitance of the sample to an optimum temperature and then decreasing because this may decrease in the specific surface area directly impact specific capacitance due to high decomposition of material after optimum temperature [34]. These results show that AC-700 °C and NAC-700 °C exhibited the best performance and it is noteworthy, from Table 1 as the L$_c$ decreases there is an increase in SSA, so it leads to increase in specific capacitance and porosity [35,36].

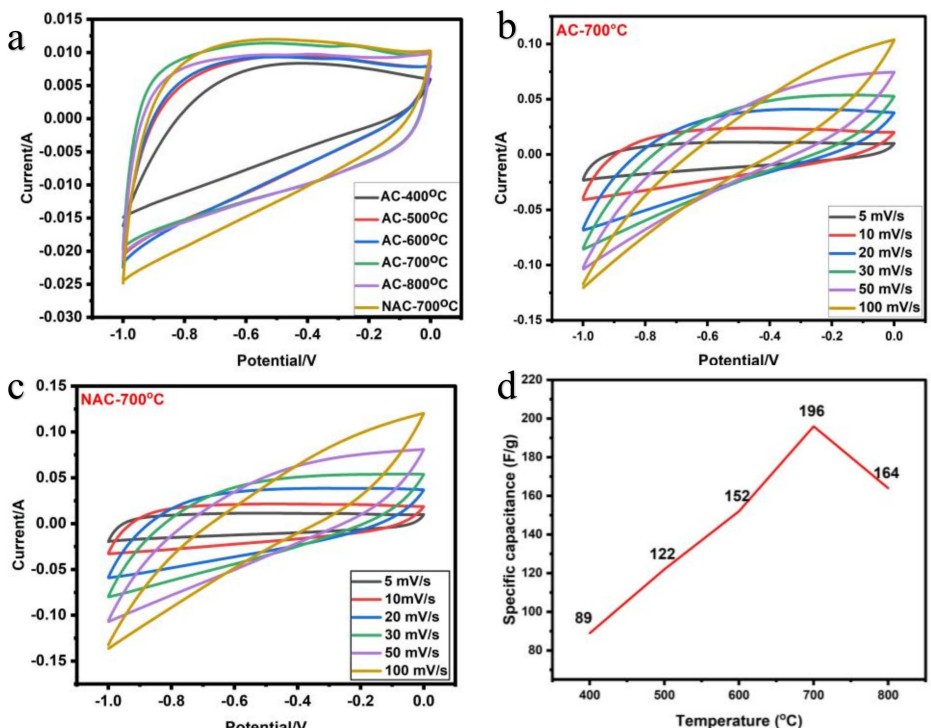

**Figure 5.** CV graph AC an NAC at various temperature (**a**) CV of AC and NAC at scan rate of 5 mV s$^{-1}$, (**b**) CV curve of AC-700 °C, (**c**) CV curve of NAC-700 °C, (**d**) Effect of temperature on specific capacitance from CV.

### 3.6. Galvano-Static Charge and Discharge (GCD)

The GCD curves have been performed for each sample (AC-400 °C to 800 °C and NAC-700 °C). Figure 6c,d shows the GCD curves for AC-700 °C and NAC-700 °C of supercapacitor devices with potential windows from −1.0 to 0 V at different current densities. The AC-700 °C and NAC-700 °C show the best electrochemical measurements compared to all other carbonization temperatures. Figure 6a shows GCD of AC and NAC at a current density of 0.25 A g$^{-1}$, It is note worthy that NAC shows a good dicscharge time. Figure 6d shows the specific capacitance of AC's is increased as with the carbonization temperature reaches up to 700 °C then start decreasing further due to the temperature rise leading to dissociation of the pore already existing [37]. Form Figure 6b,c AC-700 °C exhibits the specific capacitance of 183 F/g among all other carbonization temperature from the GCD curve and at this optimum carbonization temperature as prepared N-doped activated carbon (NAC-700 °C) shows a better specific capacitance of 258 F/g. After N-doping the improvement of specific capacitance is approximately 32%. The charge and discharge time decreases as the current densities increases because at low current density the charge transfer between the electron and electrolyte on the electrode surfaces is more [37]. The specific capacitance of all samples decreases as the current density increases. But at the same time, increase in current density didn't result in a change in the profile of the GCD graphs, revealing that both electrode material has high-rate capability. Thus, the two samples resulted in symmetric triangular shape. This quasi-symmetric triangular curve indicates the ideal properties of EDLC, high efficiency, and good reversibility in the charge-discharge process [23,38]. For comparative studies, the NAC performance as electrode materials was analyzed with other N-doped activated carbon as listed in Table 2.

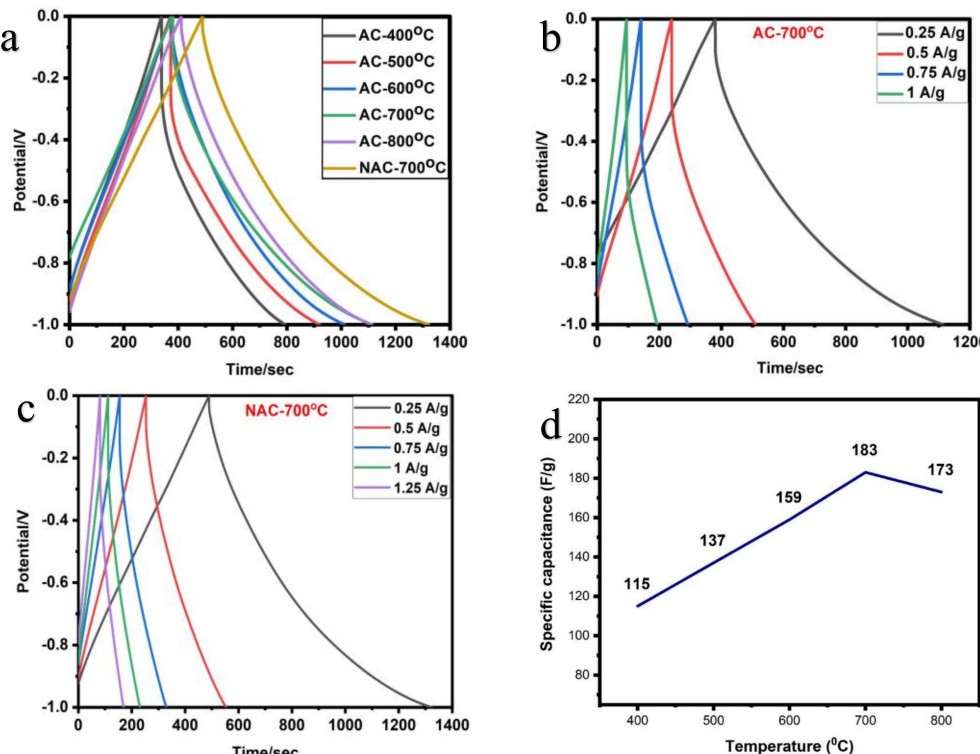

**Figure 6.** GCD graph of AC an NAC at various temperature (**a**) GCD of AC and NAC at a current density of 0.25 A g$^{-1}$, (**b**) GCD curve of AC-700 °C, (**c**) GCD curve of NAC-700 °C, (**d**) effect of temperature on specific capacitance from GCD.

**Table 2.** Comparison of electrochemical performance NAC derived from different precursors.

| Raw Materials | Precursors | Electrolyte | Specific Capacitance (F g$^{-1}$) | Energy Density (Wh kg$^{-1}$) | Ref. |
|---|---|---|---|---|---|
| Orange peel | KOH + Melamine | 6 M KOH | 168 | 23.3 | [39] |
| corncob | KOH + NH$_3$ | Organic | 185 | - | [40] |
| Pea skin | KOH + Melamine | 1 M LiTFSI in 1 L EMITFSI | 141 | 19.6 | [41] |
| Peony pollen | KOH + NH$_4$[BF$_4$] | 6 M KOH | 209 | - | [42] |
| Macadamia nutshell | KOH + Melamine | 1 M KOH | 229 | - | [43] |
| Glucosamine | - | PVA/ KOH | 244 | 7.2 | [44] |
| Tea seed shell | KOH + Melamine | 1 M KOH | 141 | - | [45] |
| Pueraria | Melamine + K$_2$CO$_3$ | 6 M KOH | 250 | 8.46 | [46] |
| Eucalyptus Leaves | ZnCl$_2$ + Urea | 2 KOH | 258 | 28.76 | Present work |

## 4. Conclusions

In summary, activated carbon (AC) and N-Doped activated carbon (NAC) were synthesized from eucalyptus leaves using chemical activation techniques at various temperature with ZnCl$_2$ as activator, and the source of nitrogen was loaded using urea. The synthesized AC at 700 °C showed a better performance with a surface area of 1027 m$^2$ g$^{-1}$ and the enhanced performance was observed in NAC at 700 °C with a surface area of have a surface area 1042 m$^2$ g$^{-1}$. The impact of activator and nitrogen doped showed different morphology such as porous, graphitic sheet, honeycomb structure. The specific capacitance of AC and NAC were obtained 194 F g$^{-1}$ and 258 F g$^{-1}$, respectively, at a scan rate of 5 mV s$^{-1}$. The energy densities of AC and NAC were obtained 22 Wh Kg$^{-1}$ and 28.76 Wh

Kg$^{-1}$, respectively, at a current density of 0.25 A g$^{-1}$, for the carbon electrode. This carbon electrode shows good performance with alkaline electrolyte. The electrochemical performance suggest that AC and NAC were potentially excellent electrode for supercapacitor cell application. Finally, the findings of this study highlight the need of defining abundant residue as a necessary step in determining prospective usage for a subsequent valorization.

**Supplementary Materials:** The following supporting information can be downloaded at: https://www.mdpi.com/article/10.3390/c9010024/s1, Figure S1: EDS spectra; Table S1: EDS summary.

**Author Contributions:** D.B.: Experimental, results, investigation, writing original draft, review, and revisions. P.B.: Experimental, results analysis, revision, and editing. V.B.K.: conceptualization review and editing. P.S.K.: Supervision; conceptualization, review, suggestions, and editing. All authors have read and agreed to the published version of the manuscript.

**Funding:** This research received no external funding.

**Data Availability Statement:** Not applicable.

**Conflicts of Interest:** The authors declare no conflict of interest.

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
