# Peer review of "Biomass-Derived N-Doped Activated Carbon from Eucalyptus Leaves as an Efficient Supercapacitor Electrode Material"

_carbon, 2022_

Round 1

Reviewer 1 Report

The article is about biomass-derived N-doped activated carbon from eucalyptus leaves as an efficient supercapacitor electrode material. However, some issues must to be addressed:

  1. Abstract: Please start by expressing the aim of this paper, followed by the rest of the information. Also, please define or try to avoid using abbreviations in the abstract. Typically, the abstract should provide a broad overview of the entire project, summarize the results, and present the implications of the research or what it adds to its field.
  2. The results are merely presented, not properly discussed. Please add explanations for the observed changes. Please give an extended discussion on the obtained results and correlate your findings with previous literature studies and prospective applications.
  3. More analysis and interpretation of the results should be added for a clearer understanding of observed experimental phenomena.
  4. The authors must to provide some details about importance of the research and their applicability.
  5. Please rewrite the conclusions in a more quantitative form and enhance the clarity of the conclusion section in order to highlight the results obtained.
  6. General check-up and correction of the English language is suggested. There are still some minor typos and grammatical errors.

The author needs to address the abovementioned points for the betterment of the manuscript.

Author Response

  1. Abstract: Please start by expressing the aim of this paper followed by the rest of information. Also, please define or try to avoid using abbreviations in the abstract. Typically, the abstract should provide a broad overview of the entire project summarize the results and present the implications of the research or what it adds to its field.

Response: In response to the reviewer's comments, the abstract has been revised and modified accordingly.

  1. The results are merely presented not properly discussed. Please ass explanations for the observed changes. Please give an extended discussion on the obtained results and correlate your findings with previous literature studies and prospective applications.

Response:  We have revised and corrected the results. Table 2 shows a correlation between the results of the present study and the prior literature.

  1. More analysis and interpretation of the results should be added for a clearer understanding of observed experimental phenomena.

Response:  According to the reviewer's suggestion, energy dispersive spectroscopy (EDS) analysis was performed for all samples.

EDS (energy dispersive spectroscopy) was used to determine the chemical composition of ACs at different carbonization temperatures as well as the composition of NAC. The chemical composition of the different samples is shown in Table S1. According to Figure S1 and Table S1, as the carbonization temperature increases, the weight and atomic percentage of carbon increase. Carbon content increased from 80.2 to 97.5 wt.% in the AC; however, oxygen content decreased from 15.0 to 0 wt.%.

Fig. 1 EDAX analysis of (a) AC-400°C, (b) AC-500°C, (c) AC-600°C, (d) AC-700°C, (e) AC-800°C, and (f) NAC-700°C.

Table S1: EDS analysis of AC at various carbonization temperatures and NAC-700°C

Elements

Samples

AC-400

AC-500

AC-600

AC-700

AC-800

NAC-700

Weight (%)

Atom (%)

Weight (%)

Atom (%)

Weight (%)

Atom (%)

Weight (%)

Atom (%)

Weight (%)

Atom (%)

Weight (%)

Atom (%)

Carbon ( C)

80.62

86.69

84.86

89.15

90.15

97.22

96.57

98.95

97.52

99.2

66.04

71.26

Nitrogen (N)

-

-

-

-

-

-

-

-

-

-

15.78

14.6

Oxygen (O)

15.06

12.15

13.01

10.26

-

-

-

-

-

-

17.08

13.83

Sulfur  (S)

0.14

0.06

0.1

0.04

-

-

0.66

0.25

0.26

0.1

0.07

0.03

Chlorine (Cl)

1.66

0.61

1

0.35

4.95

1.81

1.74

0.6

1.81

0.62

0.45

0.16

Zinc (Zn)

2.52

0.5

1.03

0.2

4.9

0.97

1.03

0.19

0.41

0.08

0.58

0.12

  1. The authors must to provide some details about importance of the research and their applicability.

Response:    “In this work, the synthesised waste biomass carbon using carbonation, studied the change in physical properties on varying temperature and also an optimum temperature has been found for better electrochemical performance”

The applicability of the research work as given below.

Functioning of Biomass derived carbon: The wettability and modifying the electrical properties with dopant such as sulfur (s), phosphorus (P) and Boron (B). The dopants aids in fast reaction with surface function group and can improve the capacitance of the activated carbon.

Enhance the energy density: The present research work demonstrated a reasonable energy density 22 Wh kg-1, which is not feasible for commercial use. In order to increase the energy density of the carbon material, there is need to synthesis composite with metal oxide and biomass derived carbon. The synergetic reaction between biomass derived carbon and metal oxides assist high capacitance, enhance in energy density and good rate capability.

Choice of Electrolyte: Electrolytes also have a significant effect on electrode performance study on various type of electrolytes such as alkaline, acid and organic electrolyte. Further studies as to carry on electrolytes.

  1. Please rewrite the conclusion in a more quantitative form and enhance the clarity of the conclusion section in order to highlight the results obtained.

Response:   The conclusion explains the results in detail and highlights them.

  1. General check-up and correction of the English language is suggested. There are still some minor typos and grammatical errors.

Response:  In response to the reviewers' suggestions, the manuscript has been thoroughly revised and the majority of typos and grammatical errors have been corrected.

Reviewer 2 Report

This article is devoted to the topic of obtaining materials for supercapacitor electrodes from biomass. The authors used Eucalyptus biomass for this purpose. In general, the article is well written and understandable. The results are undeniable and well structured. There are some points that would like to be improved:

1. Carbonation. It is desirable to justify why the authors chose such a temperature and heating rate.

2. Do macropores, micropores or mesopores predominate in the resulting carbon material? This is also important in the further use of this material.

3. Please cite: 10.1016/j.biombioe.2020.105759.

4. You can expand the description of paragraph "3.2"

5. Conclusions need significant expansion.

Author Response

  1. It is desirable to justify why the authors chose such a temperature and heating rate.

Response:  As the heating rate and carbonization temperature increases then the yields of biochar decreases. At low heating rate the yield is high (w.r.t to ref).  In the present work the experiment were carried out from low too high, for a high surface are and optimum temperature. Therefore at 700 °c optimum temperature was observed with better performance.

http://dx.doi.org/10.1016/j.biortech.2012.10.150

  1. Do macrospores, microspores or mesoporous predominate in the resulting carbon material? This is also important in the further use of this material.

Response: Thank you for pointing out the comment "In SEM images, the carbon has a good porosity with macro pores in nature and that explains the formation of the high surface area ”.

  1. Please cite: 10.1016/j.biombioe.2020.105759

Response: The reference article has been cited.

  1. You can expand the description of paragraph “3.2”.

Response: It has been explained and expanded in Section 3.2

  1. Conclusions need significant expression.

Response:   In response to the reviewer's suggestions, "the conclusion has been explained in detail and the results have been highlighted”.

Round 2

Reviewer 1 Report

The article can be published.